# Antiviral Effect of Ephedrine Alkaloids-Free Ephedra Herb Extract against SARS-CoV-2 In Vitro

**DOI:** 10.3390/microorganisms11020534

**Published:** 2023-02-20

**Authors:** Masashi Uema, Masashi Hyuga, Kenzo Yonemitsu, Sumiko Hyuga, Yoshiaki Amakura, Nahoko Uchiyama, Kazushige Mizoguchi, Hiroshi Odaguchi, Yukihiro Goda

**Affiliations:** 1National Institute of Health Sciences, 3-25-26 Tonomachi, Kawasaki-ku, Kawasaki 210-9501, Kanagawa, Japan; 2National Institute of Infectious Diseases, 4-7-1 Gakuen, Musashimurayama 208-0011, Tokyo, Japan; 3Oriental Medicine Research Center, Kitasato University, 5–9-1 Shirokane, Minato-ku, Tokyo 108-8642, Japan; 4College of Pharmaceutical Sciences, Matsuyama University, 4-2 Bunkyo-cho, Matsuyama 790-8578, Ehime, Japan; 5Kampo Research & Development Division, Tsumura & Co., 3586 Yoshiwara, Ami-machi, Inashiki-gun 300-1192, Ibaraki, Japan

**Keywords:** ephedrine alkaloids-free Ephedra Herb extract, Ephedra Herb macromolecule condensed–tannin, COVID-19, SARS-CoV-2, antiviral therapeutic

## Abstract

We report for the first time that ephedrine alkaloids-free Ephedra Herb extract (EFE) directly inhibits the replication of severe acute respiratory syndrome coronavirus 2 (SARS-CoV-2) in vitro and that the addition of EFE to the culture medium before viral infection reduces virus titers in the culture supernatant of SARS-CoV-2, including those of variant strains, by more than 99%, 24 h after infection. The addition of Ephedra Herb macromolecule condensed-tannin, which is the main active ingredient responsible for the anticancer, pain suppression, and anti-influenza effects of EFE, similarly suppressed virus production in the culture supernatant by 99% before infection and by more than 90% after infection. Since EFE does not have the side effects caused by ephedrine alkaloids, such as hypertension, palpitations, and insomnia, our results showed the potential of EFE as a safe therapeutic agent against coronavirus disease 2019.

## 1. Introduction

Coronavirus disease 2019 (COVID-19), caused by severe acute respiratory syndrome coronavirus 2 (SARS-CoV-2), has considerably affected global public health. As of 23 December 2022, 651 million confirmed cases worldwide, including 6.65 million deaths, were reported by the World Health Organization [1]. Over the past three years, research on several drugs, including antivirals and vaccines, against COVID-19 has been conducted [2]. The first therapeutic strategy against COVID-19 was to control virus proliferation and immune responses in patients at the early stages of infection to prevent severe disease [3].

Ephedra Herb is the most popular and important crude drug used in Kampo medicine, which is a Japanese traditional medicine. There are 148 ethical prescriptions that combine various natural medicines, some of which are used to treat common cold. Ephedra Herb has been reported to have anti-influenza activity [4]. Some Kampo formulas containing Ephedra Herb are used in the initial treatment of viral infections. However, Ephedra Herb has some adverse effects, such as agitation, palpitations, elevated blood pressure, insomnia, and dysuria derived from ephedrine alkaloids. We developed an ephedrine alkaloids-free Ephedra Herb extract (EFE) to relieve these adverse effects. EFE retained the same level of anti-influenza, analgesic, and anticancer effects as the Ephedra Herb [5]. We have previously reported that alkaloid-derived side effects are reduced [6], and that EFE is safer than the Ephedra Herb in humans [7].

The analgesic, anticancer, and anti-influenza effects of EFE were found to be caused by high-molecular mass condensed tannins with a molecular weight of 45,000–100,000, which are present in approximately 20% of EFE [8], designated as Ephedra Herb macromolecule condensed–tannin (EMCT).

Given that Ephedra Herb is prescribed for common cold symptoms and EFE and EMCT exhibit anti-influenza activity, this study aimed to examine the effects of EFE and EMCT on the growth of novel coronaviruses in vitro. This study also aims to provide valuable insights into the antiviral properties of EFE and EMCT to provide a foundation for its clinical application as a potential treatment for COVID-19 patients.

## 2. Materials and Methods

### 2.1. Cells and Viruses

VeroE6/TMPRSS2 cells expressing the transmembrane serine protease 2 gene and showing high susceptibility to SARS-CoV-2 [9] were obtained from the Japanese Collection of Research Bioresources Cell Bank and cultured in Dulbecco’s modified Eagle’s medium (DMEM, Wako Pure Chemicals, Tokyo, Japan) supplemented with 5% fetal bovine serum (FBS) and penicillin-streptomycin. SARS-CoV-2/WK-521 [9], QK002 and QHN001 (alpha variant), TY7-501 and TY7-503 (beta variant), TY8-612 (gamma variant), TY38-873 (BA.1 Omicron variant), and TY41-702 (BA.5 Omicron variant) were obtained from the National Institute of Infectious Diseases, Japan. The virus was propagated in VeroE6/TMPRSS2 cells, maintained in DMEM supplemented with 5% FBS, and stored at −80 °C until use. All experiments were performed at the Biosafety Level 3 Laboratory of the National Institute of Health Sciences.

### 2.2. Preparation of EFE and EMCT

The preparation of EFE was carried out as described previously [10]. Briefly, Ephedra Herb (*E. sinica*, the Japanese pharmacopoeia grade) was added to water, extracted at 95 °C for 1 h, and filtered, after which the residue was washed with water. The extract was centrifuged at 1800× *g* for 10 min, and then the supernatant was passed directly through the DIAION SK-1B ion-exchange resin. The unadsorbed fraction was adjusted to pH 5 using 5% NaHCO_3_ aq., and the solution was then evaporated under reduced pressure to obtain EFE. 

EMCT was prepared according to previous reports [8]. Briefly, Ephedra Herb extract was dissolved in H_2_O and extracted with ethyl acetate and *n*-butanol. The H_2_O extract obtained was separated using column chromatography over DIAION HP-20 with methanol (MeOH)-H_2_O (0:100→100:0) in stepwise grade mode. Then, 40% MeOH eluate was separated using column chromatography on Sephadex LH-20 with MeOH-H_2_O (50:50→80:20) and 70% acetone in a stepwise-gradient manner to obtain a 70% acetone fraction as the EMCT fraction. The confirmation of EMCT was performed by gel permeation chromatography [8]. EFE (10 mg/mL) and EMCT (1 mg/mL) were dissolved in ultra-pure water and stored at –80 °C until further use.

### 2.3. Virus Growth Inhibition Assay

A day before the experiment, the cells were seeded in 96-well plates at 20,000 cells/well and incubated overnight at 37 °C. For the pretreatment experiment, the medium was changed to an EFE- or EMCT-containing medium 2 h before virus inoculation, and then the virus was inoculated at multiplicity of infection =0.03 and incubated for 2 h. The medium was then replaced with a new medium containing EFE or EMCT and incubated for 24 h. At 24 h post-infection, a series of 10-fold dilutions of the collected culture supernatant from each well were prepared and inoculated into cells seeded in 96-well plates. Plates were incubated for four days at 37 ℃, and the 50% tissue culture infectious dose (TCID_50_) was determined. 

In the post-treatment experiments, the medium was changed to a new medium 2 h prior to virus infection, and virus inoculation and culture incubation were performed as described in the pretreatment experiments.

At least two independent experiments were conducted and the number of replicates was indicated in the tables.

### 2.4. TCID_50_ Assay

VeroE6/TMPRSS2 cells were seeded in 96 well plate one day before the experiment and incubated at 37 °C. Next, 50 μL of the 10-fold serially diluted supernatant of each sample treated with EFE or EMCT was added, and the plates were incubated at 37 °C for four days. The cytopathic effect was monitored, and the viral titer was determined. 

### 2.5. Cell Viability Assay

A series of 2-fold dilution starting at 320 µg/mL for EFE and 80 µg/mL for EMCT was prepared. Cells seeded in 96-well plates were incubated with a medium containing EFE or EMCT for 24 h, and the cells that survived were measured using the Cell Counting Kit-8 (Dojindo, Kumamoto, Japan) following the manufacturer’s instructions. Two independent experiments were conducted.

### 2.6. Statistical Analysis

All data are expressed as the mean ±standard deviation. Significant differences between the control and treatment groups were determined by one-way ANOVA and Dunett’s test using EZR [11]. Statistical significance was set at *p* < 0.05.

## 3. Results

### 3.1. Cell Viability under EFE or EMCT Treatment

Cell viability after 24 h of incubation with EFE or EMCT was shown in Figure 1. At a concentration of 80 µg/mL, the EFE and EMCT treatment groups showed 89.2 and 92.7% viability, respectively, when compared to the control, indicating that EFE and EMCT showed low toxicity to the VeroE6/TMPRSS2 cells.

### 3.2. Virus Growth Inhibition of Pretreatment with EFE or EMCT

First, we investigated the effect of EFE addition on viral growth using the WK-521 strain. When EFE was added up to 100 µg/mL 2 h before virus infection, the log_10_ reduction in virus titer in the culture supernatant at 24 h post-infection at EFE concentrations of 3.13, 6.25, 12.5, 25.0, 50.0, and 100.0 µg/mL, were 0.63, 0.82, 1.00, 0.91, 1.66, and >5, respectively (Table 1, Figure 2A). 

Second, as it has been suggested that the anti-influenza effects of EFE are due to the presence of approximately 20% EMCT, we investigated the inhibitory effect of EMCT against virus growth. When EMCT was added 2 h before viral infection, the log_10_ reduction in virus titer in the culture supernatant at 24 h post-infection at EMCT concentration of 2.5, 5.0, 10.0, 20.0, 30.0, and 40.0 µg/mL, were −0.09, −0.28, 0.53, 2.03, >3.91, and >3.91, respectively (Table 1, Figure 2A). These results showed a concentration-dependent decrease in infectious viruses in the culture supernatant.

### 3.3. Virus Growth Inhibition of Post-Treatment with EFE or EMCT

When EFE was added up to 100 µg/mL 2 h after virus infection, the log_10_ reduction in virus titer in the culture supernatant at 24 h post-infection at EFE concentrations of 3.13, 6.25, 12.5, 25.0, 50.0, and 100.0 µg/mL were 0.78, 0.91, 1.08, 0.87, 1.58, and 2.12, respectively (Table 2, Figure 2B). 

When EMCT was added 2 h after viral infection, the log_10_ reduction in virus titer in the culture supernatant at 24 h post-infection at EMCT concentrations of 3.13, 6.25, 12.5, 25.0, and 50.0 µg/mL, were 0.03, 0.10, 0.53, 2.41, and 3.60, respectively (Table 2, Figure 2B). 

### 3.4. Inhibitory Effect of EFE or EMCT against Variant Strains

Given that the global infection situation has changed dramatically with the advent of variants of concern, we examined the effects of EFE and EMCT on the growth of alpha, beta, gamma, and omicron variant strains.

Following the results for the WK-521 strain, when EFE was added 2 h before infection at a concentration of 100 µg/mL, the log_10_ reductions in QK002 (alpha), QHN001 (alpha), TY7-501 (beta), TY7-503 (beta), TY8-612 (gamma), TY38-873 (omicron BA.1), and TY41-702 (omicron BA.5) virus titers in the culture supernatant at 24 h post-infection were 3.06, >5, >4.5, 4.25, >4.5, 2.25, and 2.04, respectively (Table 3, Figure 3A). When EMCT was added prior to infection at a concentration of 25 µg/mL, the log_10_ reduction in QK002 (alpha), QHN001 (alpha), TY7-501 (beta), TY7-503 (beta), TY8-612 (gamma), TY38-873 (omicron BA.1), and TY41-702 (omicron BA.5) virus titers in the culture supernatant at 24 h post-infection were 2.81, >5, >4.5, >4.6, >4.5, 2.73, and 2.91, respectively (Table 3, Figure 3B).

When EFE was added 2 h after infection at a concentration of 100 µg/mL, the log_10_ reductions in QK002 (alpha), QHN001 (alpha), TY7-501 (beta), TY7-503 (beta), TY8-612 (gamma), TY38-873 (omicron BA.1), and TY41-702 (omicron BA.5) virus titers in the culture supernatant at 24 h post-infection were 1.31, 2.19, 1.06, 1.0, 1.56, 2.27, and 2.46, respectively (Table 3, Figure 2B). When EMCT was added after infection at a concentration of 25 µg/mL, the log_10_ reduction in QK002 (alpha), QHN001 (alpha), TY7-501 (beta), TY7-503 (beta), TY8-612 (gamma), TY38-873 (omicron BA.1), and TY41-702 (omicron BA.5) virus titers in the culture supernatant at 24 h post-infection were 1.06, 1.37, 2.12, 1.75, 1.93, 1.97, and 2.29, respectively (Table 3, Figure 3B).

## 4. Discussion

It has been indicated that EFE has an inhibitory effect on the growth of novel coronaviruses and that its activity is derived from EMCT. In addition to VeroE6/TMPRSS2 cells, A549, Calu-3, and other cells have been used in novel coronavirus studies. However, since our purpose was to compare the inhibition effects of EFE and EMCT on variant strains in parallel, including the Omicron-type, VeroE6/TMPRSS2 cells were suitable. Under EFE and EMCT treatment conditions, the reduction in infectious viral load was observed within the culture supernatant after 24 h incubation. Regarding the mechanism of inhibition, it is suggested that EMCT competitively binds to the receptor-binding domain on the spike protein of SARS-CoV-2 with angiotensin-converting enzyme 2 [unpublished data], suggesting that viral entry into cells via the receptor pathway was inhibited by EMCT. 

The addition of 100 µg/mL of EFE and 25 µg/mL of EMCT suppressed the growth of infectious viruses in the culture supernatant at 24 h post-infection to the same extent. This result is consistent with a report that the viability of influenza virus-infected MDCK cells 72 h post-infection was similar to when 100 µg/mL of EFE or 25 µg/mL of EMCT was added [5]. Although the growth of the WK-521 strain was not inhibited at low EMCT concentrations of 2.5 and 5 µg/mL, respectively, during pretreatment (Table 1, Figure 2A) and at 3.13 and 6.25 µg/mL, respectively, during post-treatment (Table 2, Figure 2B), the inhibition of WK-521 growth was observed even at low concentrations of 3.13 and 6.25 µg/mL of EFE, respectively (Table 1 and Table 2, Figure 2). Treatment with EMCT at low concentrations was insufficient to inhibit viral entry into cells, suggesting that factors other than EMCT contained in EFE may also inhibit coronavirus growth.

In the present study, EFE treatment prior to viral infection showed a greater reduction in virus growth, indicating the importance of inhibiting the initial cell invasion stage in the life cycle of novel coronaviruses. However, since the titer of infectious virus in the culture supernatant was reduced by more than 90%, even when EFE was added after infection, it indicated that the suppression of viral transmission occurred after the initial viral entry step.

In the case of the omicron-type variants, EFE pretreatment resulted in a smaller reduction in infectious virus titer after 24 h than those of the alpha, beta, and gamma viruses, whereas post-infection treatment resulted in a reduction similar to other variants. Several reports suggest that the omicron-type variants infect culture cells mainly by using the cathepsin-dependent endocytosis pathway, and use the furin/TMPRSS2-dependent pathway inefficiently [12], and that they are less fusogenic than the Delta and ancestral strains [13]. In the experiments using omicron variants, these properties may be the reason why the pre-treatment with EFE or EMCT resulted in higher viral titers at 24 h when compared to other variants, while the differences in the reduction in viral load between post- and pre-treatment were small. Furthermore, higher infectivity and numerous amino acid changes in the spike proteins of the omicron strain when compared to those seen in pre-delta variants have been suggested [14,15]. It is possible that the inhibition of the cell entry process, facilitated by the binding of EMCT and the spike protein of omicron variants, may be weak. Further studies on the mechanism of antiviral action of EFE and EMCTs constitute the basis for future work. As the study was only able to investigate a small number of clinical isolates, it is difficult to determine whether the efficacy of EFE and EMCT varies by variant type or whether there is a trend; however, the fact that the two alpha-type isolates showed differences in the suppression of viral replication by EFE and EMCT suggests that there exists a difference between isolates.

EFE is a crude drug, different from existing antiviral drugs, and it exhibits various medicinal effects, such as analgesic and anticancer effects at the cellular and individual animal levels, as previously reported [5]. In particular, the anticancer activity of EFE and EMCT displays kinase inhibitor activity that prevents c-Met phosphorylation, and their growth inhibitory effect against novel coronaviruses indicates the possibility of inhibiting several steps of the viral life cycle, in addition to the cell entry stage [16,17].

In addition to the direct binding between viral spike proteins and EFE or EMCT, the medical effects at the cellular and physiological levels are expected to lead to an improvement in the general condition of infected patients. Unlike existing antiviral therapeutics, which consist of a single compound, this drug has high potential as a new therapeutic agent with many points of action against viral infections. 

The deduced structures of the active components of EMCT were found to be high-molecular mass condensed tannins, which were primarily B-type procyanidin and partly A-type procyanidin, including pyrogallol- and catechol-type flavan 3-ols as extension and terminal units, respectively. High-performance liquid chromatography and gel permeation chromatography analyses estimated that the ratio of pyrogallol- to catechol-type was approximately 9:2, and the weight-average molecular weight based on the polystyrene standard was >45,000. [8]. A-type procyanidin is expected to have a planar structure, possibly forming binding domains with the proteins. The possibility of multiple binding domains within a single EMCT molecule and the possibility of different properties at different molecular weights were suggested.

Ephedra Herb is the most popular and important ingredient among the Kampo formulas prescribed for viral infections such as the common cold, and EMCT, which is a very large molecular tannin, and is considered to be the main antiviral agent of EFE. Therefore, EFE may bind to a variety of proteins and has potential as a therapeutic agent with antiviral activity against viruses other than the novel coronaviruses and influenza viruses.

EFE has been shown to reduce the side effects of Ephedra Herb such as excitation, insomnia, and arrhythmias [6], and to have a high safety profile for humans [7]. Thus, it is expected to have a wide range of applications for the treatment of COVID-19.

Although EFE and EMCT have been reported to have an indirect antiviral effect by increasing the survival rate of MDCK cells after influenza virus infection [5], this is the first study to show that EFE or EMCT directly inhibits the growth of infectious SARS-CoV-2 in vitro.

## Figures and Tables

**Figure 1 microorganisms-11-00534-f001:**
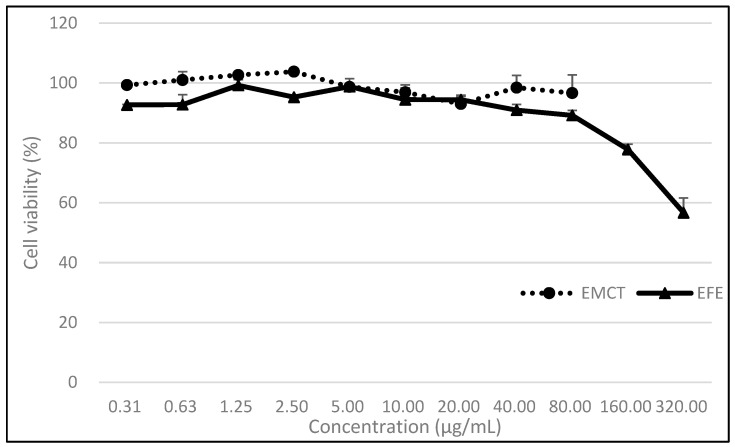
Cell viability under EFE or EMCT treatment. Seeded cells were treated with EFE or EMCT at the indicated concentrations and incubated for 24 h. Cell counts were compared to those of the control (0 µg/mL) and % viabilities are mean ± standard deviations of two independent experiments.

**Figure 2 microorganisms-11-00534-f002:**
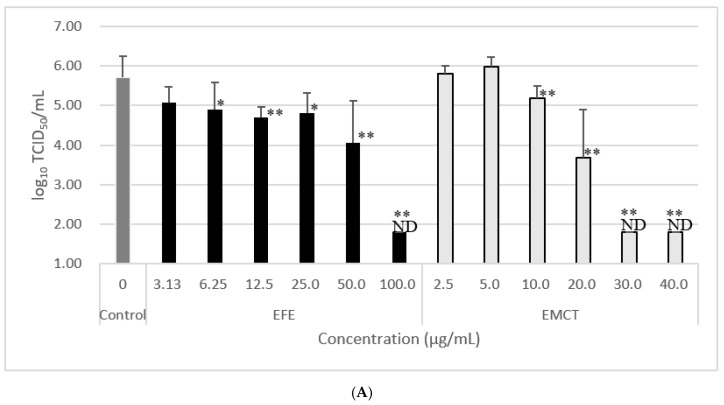
Effect of EFE or EMCT treatment against the growth of SARS-CoV-2 WK-521 strain in vitro at 24 h post-infection. The titer of virus within culture supernatant decreased in a dose-dependent manner. (**A**) EFE or EMCT was added 2 h before virus inoculation. (**B**) EFE or EMCT was added 2 h after virus inoculation. Data are presented as mean ± standard deviation. ND, not detected. Significant differences as compared with control (0 µg/mL) *; *p* < 0.05. **; *p* < 0.01.

**Figure 3 microorganisms-11-00534-f003:**
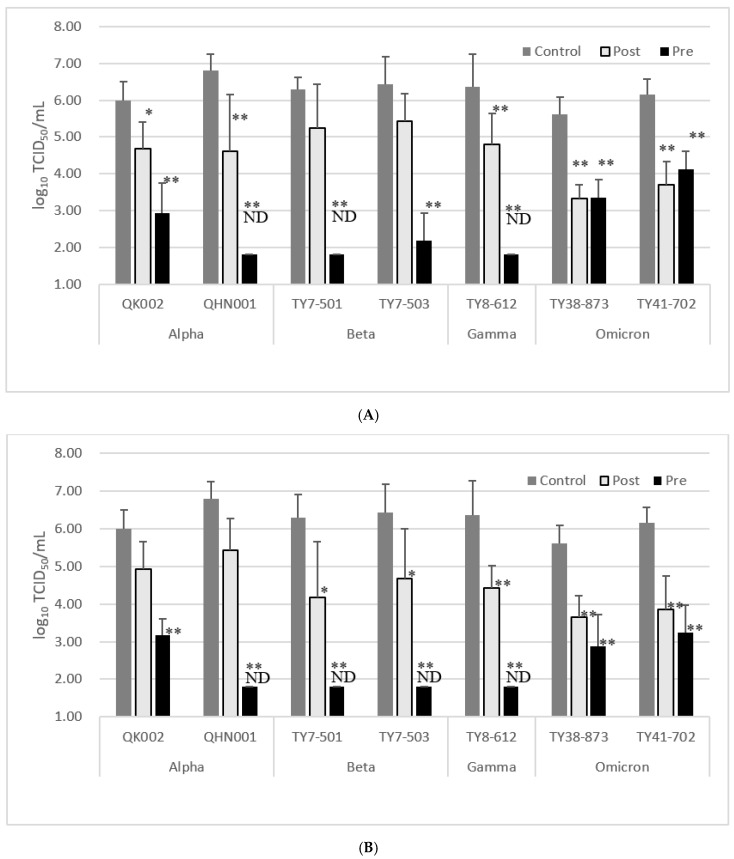
Effect of EFE (100 µg/mL) or EMCT (25 µg/mL) treatment against the growth of SARS-CoV-2 variant strains in vitro at 24 h post-infection. Pre; EFE or EMCT was added 2 h before virus inoculation. Post; EFE or EMCT was added 2 h after virus inoculation. Data are presented as mean ± standard deviation. Limit of detection of log_10_TCID_50_/mL is 1.8. (**A**) Effect of EFE treatment against variant strains. (**B**) Effect of EMCT treatment against variant strains. ND, not detected. Significant differences as compared with control (0 µg/mL) *; *p* < 0.05. **; *p* < 0.01.

**Table 1 microorganisms-11-00534-t001:** Reduction of log10TCID50 of SARS-CoV-2(WK-521 strain) within culture supernatant at 24 h post infection treatment with EFE or EMCT before infection.

EFE Concentration (µg/mL)	Log10TCID50/mL	Reduction of Log10tcid50.mL		EMCT Concentration (µg/mL)	Log10TCID50/mL	Reduction of Log10tcid50.mL	
control (0)	5.71 ± 0.55						
100.00	ND	>3.91	** 1	40	ND	>3.91	** 1
50.00	4.05 ± 1.07	1.66	** 1	30	ND	>3.91	** 1
25.00	4.80 ± 0.52	0.91	* 1	20	3.68 ± 1.23	2.03	** 2
12.50	4.71 ± 0.27	1.00	** 1	10	5.18 ± 0.32	0.53	** 2
6.25	4.89 ± 0.68	0.82	* 1	5	5.99 ± 0.24	−0.28	3
3.13	5.08 ± 0.39	0.63	1	2.5	5.8 ± 0.20	−0.09	3

ND = not detected, Limit of TCID50/mL = 1.8, Data are presented as mean ± standard deviation of 1: four, 2: six, 3: two independent experiments. Significant differences as compared with control (0 µg/mL) (* *p* < 0.05, ** *p* < 0.01).

**Table 2 microorganisms-11-00534-t002:** Reducation of log10TCID50 of SARS-CoV-2(WK-521 strain) within culture supernatant at 24 h post infection treatment with EFE or EMCT after infection.

EFE Concentration (µg/mL)	Log10TCID50/mL	Reduction of LogTCID50		EMCT Concentration (mg/mL)	Log10TCID50/mL	Reduction of LogTCID50	
control (0)	5.71 ± 0.55						
100.00	3.59 ± 0.77	2.12	** 1	50.00	2.11 ± 0.55	3.60	** 2
50.00	4.13 ± 1.32	1.58	** 1	25.00	3.30 ± 1.01	2.41	** 2
25.00	4.84 ± 0.43	0.87	1	12.50	5.18 ± 0.86	0.53	2
12.50	4.63 ± 0.44	1.08	* 1	6.25	5.61 ± 0.76	0.10	2
6.25	4.80 ± 0.27	0.91	1	3.13	5.68 ± 0.68	0.03	2
3.13	4.93 ± 0.63	0.78	1				

Data are presented as mean ± standard deviation of 1: three, 2: four independent experiments. Significant differences as compared with control (0 µg/mL) (* *p* < 0.05, ** *p* < 0.01).

**Table 3 microorganisms-11-00534-t003:** Reducation of log10TCID50 of variant strains within culture supernatant at 24 h post infection.

		Treatment before Infection							Treatment after Infection					
		EFE (µg/mL)				EMCT (µg/mL)			EFE (µg/mL)			EMCT (µg/mL)		
	Variant	Control (0)	100	Reduction of Log TCID50		25	Reduction of Log TCID50		100	Reduction of Log TCID50		25	Reduction of Log TCID50	
QK002	Alpha	5.99 ± 0.52	2.93 ± 0.83	3.06	** 1	3.18 ± 0.43	2.81	** 1	4.68 ± 0.72	1.31	* 1	4.93 ± 0.72	1.06	1
QHN001	Alpha	6.8 ± 0.46	ND	>5	** 1	ND	>5	** 1	4.61 ± 1.53	2.19	** 1	5.43 ± 0.85	1.37	1
TY7-501	Beta	6.3 ± 0.61	ND	>4.5	** 1	ND	>4.5	** 1	5.24 ± 1.20	1.06	1	4.18 ± 1.48	2.12	* 1
TY7-503	Beta	6.43 ± 0.75	2.18 ± 0.75	4.25	** 1	ND	>4.6	** 1	5.43 ± 0.75	1	1	4.68 ± 1.33	1.75	* 1
TY8-612	Gamma	6.36 ± 0.90	ND	>4.5	** 1	ND	>4.5	** 1	4.8 ± 0.84	1.56	** 1	4.43 ± 0.60	1.93	** 1
TY38-873	Omicron BA.1	5.61 ± 0.48	3.36 ± 0.48	2.25	** 2	2.88 ± 0.84	2.73	** 2	3.34 ± 0.85	2.27	** 3	3.64 ± 0.58	1.97	** 3
TY41-702	Omicron BA.5	6.15 ± 0.42	4.11 ± 0.51	2.04	** 2	3.24 ± 0.73	2.91	** 2	3.69 ± 0.79	2.46	** 3	3.86 ± 0.88	2.29	** 3

ND = not detected, Limit of TCID50/mL = 1.8. Data are presented as mean ± standard deviation of 1: two, 2: six, 3: four independent experiments. Significant differences as compared with control (0 µg/mL) (* *p* < 0.05, ** *p* < 0.01).

## Data Availability

The data supporting the results are available upon request.

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
