# Peer review of "Antiviral Effect of Ephedrine Alkaloids-Free Ephedra Herb Extract against SARS-CoV-2 In Vitro"

_microorganisms, 2023, doi:10.3390/microorganisms11020534_

Round 1

Reviewer 1 Report

In the manuscript “Antiviral effect of ephedrine alkaloid-free Ephedra Herb extract against SARS-CoV-2 in vitro" authors M. Uema et al. are describing the antiviral effect of two types of Ephedra Herb extracts ephedrine alkaloid-free EFE and Ephedra Herb macromolecule condensed-tannin EMCT on SARS-CoV2 infection in VeroE6 cell line.

              This manuscript needs to have very serious editing of both English grammar and the correctness of scientific phrases. For example, phrase 218-226 is taking the whole paragraph and is very hard for reading. Several phrases that need to be edited: title of results 3.1 line 110;  lines 116-119; lines 202-205.

Major :

1.       My main concern is that, as mentioned by the authors, both EMCT and EFE are known for their antiproliferative effects as anti-cancer substances. It is very well known that inhibition of cell proliferation in cell culture results in the suppression of viral infection. Usually, in cells that do not have strong proliferation, those substances are not showing an efficient anti-viral effect. Due to that characteristic of the substances used in the test, authors need to show more results of the effect on the cell counts and the presence or absence of cell cycle arrest. In this type of work, the cell counts would be plotted right aside of the virus titer for each dose of the tested substance.  However, in the current version there is no data for cell counts but just mentions one number for all different tests.

2.       Of cause ideally would be to show the same anti-viral effect in any other cell line or even better primary cells.

3.       The conclusion about the difference between the effect of EMCT and EFE on Omicron vs other isolates is not well explained. It would be important to show the difference in the infectivity of each isolate and compare the length of the viral-life cycle of each isolate to prove that it is not the reason of the difference in the final viral titer.

4.       Authors describe the process of generating the EMCT and EFE extracts but show no data about its characterization  and purity and  preparation of those substances and treatment of cells. It is not clear if the controls were treated with the same amount of solvents(diluent solutions) as the test samples treated with EMCT & EFE.

5.       To make a solid conclusion about the difference in the efficacy of the anti-viral effect on Omicron vs other isolate it would be good to use more doses of substances and calculation at inhibitory concentration 50 or similar values.

Minor:

1.       The full names of EMCT and EFE are mentioned too many times after the introduction of the abbreviation. This is not correct and very strongly affects the quality of phrases. Please remove full names also from the figure legends.

2.       For all figures description need to be corrected and show the decryption of data type in the plot, what was used as error bar and how many replicates were used for statistical analysis.

3.        Also in the description of methods,  need to mention how many replicates were used for statistical analysis.

4.       Authors keep calling graphs in the figures incorrectly as “Table”, for instance, lines 192, 193, 195. Also, they need a clear label for each graph in each Figure and use that ”name” for reference. Currently, authors are using A and B for graphs in the figure description and Tables 1, 2 in the text.

Author Response

Responses to Reviewer's Comments

              We appreciate reviewer #1 and reviewer #2 for their thoughtful advice and important remarks. Following their advice, we have corrected the manuscript and indicated it in red. Two references (12 and 13) were added. Our responses to reviewers’ comments are listed below.

Dear Reviewer 1

In the manuscript “Antiviral effect of ephedrine alkaloid-free Ephedra Herb extract against SARS-CoV-2 in vitro" authors M. Uema et al. are describing the antiviral effect of two types of Ephedra Herb extracts ephedrine alkaloid-free EFE and Ephedra Herb macromolecule condensed-tannin EMCT on SARS-CoV2 infection in VeroE6 cell line.

              This manuscript needs to have very serious editing of both English grammar and the correctness of scientific phrases. For example, phrase 218-226 is taking the whole paragraph and is very hard for reading. Several phrases that need to be edited: title of results 3.1 line 110; lines 116-119; lines 202-205.

Duplicated or difficult-to-understand phrases in the discussion section (page 8, lines 239-248) have been removed and reorganized. We have conducted another English editing before resubmitting the manuscript, and we also submit a certification sheet from Editage.

Major :

  1. My main concern is that, as mentioned by the authors, both EMCT and EFE are known for their antiproliferative effects as anti-cancer substances. It is very well known that inhibition of cell proliferation in cell culture results in the suppression of viral infection. Usually, in cells that do not have strong proliferation, those substances are not showing an efficient anti-viral effect. Due to that characteristic of the substances used in the test, authors need to show more results of the effect on the cell counts and the presence or absence of cell cycle arrest. In this type of work, the cell counts would be plotted right aside of the virus titer for each dose of the tested substance.  However, in the current version there is no data for cell counts but just mentions one number for all different tests.

A1; thank you for pointing it out. The measurement of cell viability at 24 hours after the addition of EFE or EMCT is added as Figure 1 (page 3, Line 125-133). We think that at the concentrations added in our experiment, the decrease in viability was small, so the possibility of the effects of EFE or EMCT against the virus life cycle is little.

  1. Of cause ideally would be to show the same anti-viral effect in any other cell line or even better primary cells.

A2; in the discussion section (page 8, Lines 212-221), it was added that the reason why we use VeroE6/TMPRSS2 cells in our study. We understand that in addition to VeroE6/TMPRSS2 cells, such as A549 and Calu-3 and other cells have been used in SARS-CoV-2 studies. Since we aimed to compare the effects against variant strains side-by-side, we chose VeroE6/TMPRSS2 cells. We could observe that each virus strain multiplies efficiently as early as possible after infection and that adding EFE or EMCT inhibits viral multiplication 24 h after infection. For example, the Omicron-type strain, as reported by several studies, replicates weakly in Calu-3 cells, so we did not think Calu-3 cells could be used for our study, and VeroE6/TMPRSS2 cells were suitable.

  1. The conclusion about the difference between the effect of EMCT and EFE on Omicron vs other isolates is not well explained. It would be important to show the difference in the infectivity of each isolate and compare the length of the viral-life cycle of each isolate to prove that it is not the reason of the difference in the final viral titer.

A3; several reports suggest that Omicron differs from other variants in that endocytosis is a more important route of entry into cells than Furin/TMPRSS2, and that it is less able to spread via fusion to adjacent cells after infection. These properties of the Omicron type variant may be why the pre-treatment of EFE or EMCT resulted in higher viral titers at 24 h post-infection than the other strains, while the difference in reduction of viral load between post-treatment and pre-treatment was small. In addition, while other strains are thought to spread cell-cell by fusion with neighboring cells as well as shedding virus from cells after invading cells, Omicron is thought to spread infection mainly by shedding virus due to its weak fusion property, which is why post-treatment of EFE or EMCT has a more significant inhibitory effect than other variant strains. In the discussion section, we added these to page 8, lines 241-248.

  1. Authors describe the process of generating the EMCT and EFE extracts but show no data about its characterization  and purity and  preparation of those substances and treatment of cells. It is not clear if the controls were treated with the same amount of solvents(diluent solutions) as the test samples treated with EMCT & EFE.

A4; thank you for pointing it out. In the materials and methods section 2.2, Page 2, lines 73-88, purification and characterization of EFE and EMCT were added. EFE and EMCT are dissolved in purified water and stored at -80 °C until further use, diluted with medium, and added at the time of use.

  1. To make a solid conclusion about the difference in the efficacy of the anti-viral effect on Omicron vs other isolate it would be good to use more doses of substances and calculation at inhibitory concentration 50 or similar values.

A5; we understand that IC50 is a very useful index for comparing drug efficacy. In the TCID50 assay of our study, if IC50 was defined as half of the viral titer in the medium, it corresponds to a Log reduction of approximately 0.3, which is not significantly different from the control. Therefore, the Log TCID50 is statistically compared. As shown in Figure 1, it is presumed that cytotoxicity becomes more potent at higher concentrations, and it was determined that verification at higher concentrations was not possible.

Minor:

We apologize for our mistakes, and carefully revised our manuscript as pointed out by the reviewer 1.

  1. The full names of EMCT and EFE are mentioned too many times after the introduction of the abbreviation. This is not correct and very strongly affects the quality of phrases. Please remove full names also from the figure legends.

A1; we have removed the full names of EFE or EMCT.

  1. For all figures description need to be corrected and show the decryption of data type in the plot, what was used as error bar and how many replicates were used for statistical analysis.

A2; we added the description that data were presented as mean ± SD, as well as the number of replicates.

  1.     Also in the description of methods, need to mention how many replicates were used for statistical analysis.

A3; we added that at least two replicates were used to confirm the results and that the number of replicates was shown in the tables.

  1. Authors keep calling graphs in the figures incorrectly as “Table”, for instance, lines 192, 193, 195. Also, they need a clear label for each graph in each Figure and use that ”name” for reference. Currently, authors are using A and B for graphs in the figure description and Tables 1, 2 in the text.

A4; we have corrected this as per your advice. We apologize for the misunderstanding caused by submitting the tables separately from the manuscript in the first submission to prevent the table layout from being broken. We have resubmitted the manuscript, including tables and figures as one file.

Reviewer 2 Report

The Communication of "Antiviral effect of ephedrine alkaloids-free Ephedra Herb extract against SARS-CoV-2 in vitro" by Uema et al. is well written and the results support their conclusion. I think the article can be accepted in Microorganisms.

My general comment is however the authors mentioned that the detailed mechanstic studies will be conducted in future, I think it would be very helpful to add in this communication an in situ work (such as molecular docking anmolecular dynamics simulations) to examine interaction between the extracted compound and spike protein of different SARS-CoV-2 variants.

Author Response

Responses to Reviewer's Comments

              We appreciate reviewer #1 and reviewer #2 for their thoughtful advice and important remarks. Following their advice, we have corrected the manuscript and indicated it in red. Two references (12 and 13) were added. Our responses to reviewers’ comments are listed below.

Dear Reviewer 2,

The Communication of "Antiviral effect of ephedrine alkaloids-free Ephedra Herb extract against SARS-CoV-2 in vitro" by Uema et al. is well written and the results support their conclusion. I think the article can be accepted in Microorganisms.

We thank you very much for your high evaluation of our research results.

My general comment is however the authors mentioned that the detailed mechanistic studies will be conducted in future, I think it would be very helpful to add in this communication an in situ work (such as molecular docking and molecular dynamics simulations) to examine interaction between the extracted compound and spike protein of different SARS-CoV-2 variants.

A; thank you very much for your valuable suggestions. Since EMCT is not a single compound, its molecular structure is challenging to determine at present. We think that in situ simulation is also difficult to conduct. We are attempting to simulate the docking of the estimated substructure of the EMCT and the prototype S-protein, results of the simulation will be discussed in another report.

Round 2

Reviewer 1 Report

Thank you for your detailed response and corrections.

Best wishes with your further work.